# Impact of Annealing in Various Atmospheres on Characteristics of Tin-Doped Indium Oxide Layers towards Thermoelectric Applications

**DOI:** 10.3390/ma17184606

**Published:** 2024-09-20

**Authors:** Anna Kaźmierczak-Bałata, Jerzy Bodzenta, Piotr Szperlich, Marcin Jesionek, Anna Michalewicz, Alina Domanowska, Jeyanthinath Mayandi, Vishnukanthan Venkatachalapathy, Andrej Kuznetsov

**Affiliations:** 1Institute of Physics—CSE, Silesian University of Technology, Konarskiego 22B, 44-100 Gliwice, Poland; jerzy.bodzenta@polsl.pl (J.B.); piotr.szperlich@polsl.pl (P.S.); marcin.jesionek@polsl.pl (M.J.); anna.michalewicz@polsl.pl (A.M.); alina.domanowska@polsl.pl (A.D.); 2Department of Materials Science, School of Chemistry, Madurai Kamaraj University, Madurai 625 021, India; jeyanthinath.chem@mkuniversity.ac.in; 3Department of Physics, University of Oslo, P.O. Box 1048 Blindern, NO-0316 Oslo, Norway; vishnukanthan.venkatachalapathy@smn.uio.no (V.V.); andrej.kuznetsov@fys.uio.no (A.K.)

**Keywords:** indium tin oxide, thin film, post-processing, thermal conductivity, thermoelectric parameters

## Abstract

The aim of this work was to investigate the possibility of modifying the physical properties of indium tin oxide (ITO) layers by annealing them in different atmospheres and temperatures. Samples were annealed in vacuum, air, oxygen, nitrogen, carbon dioxide and a mixture of nitrogen with hydrogen (NHM) at temperatures from 200 °C to 400 °C. Annealing impact on the crystal structure, optical, electrical, thermal and thermoelectric properties was examined. It has been found from XRD measurements that for samples annealed in air, nitrogen and NHM at 400 °C, the In_2_O_3_/In_4_Sn_3_O_12_ share ratio decreased, resulting in a significant increase of the In_4_Sn_3_O_12_ phase. The annealing at the highest temperature in air and nitrogen resulted in larger grains and the mean grain size increase, while vacuum, NHM and carbon dioxide atmospheres caused the decrease in the mean grain size. The post-processing in vacuum and oxidizing atmospheres effected in a drop in optical bandgap and poor electrical properties. The carbon dioxide seems to be an optimal atmosphere to obtain good TE generator parameters—high ZT. The general conclusion is that annealing in different atmospheres allows for controlled changes in the structure and physical properties of ITO layers.

## 1. Introduction

Transparent conducting oxide materials (TCO), among them the tin-doped indium oxide (ITO), zinc oxide (ZnO) and tin oxide (SnO_2_) are already widely used in commercial thin film solar cells [1,2,3] and are promising thermoelectric materials [4]. The basic requirements for transparent conductive oxides are low electrical resistance and high transmittance in the visible light range. ITO, typically fabricated as an n-type semiconductor, shows high electrical conductivity and optical transmittance; however, the degradation of the layers was observed while exposed to oxidizing atmospheres above 300 °C, such as air and oxygen [5]. The thermal stability of ITO layers in varying conditions is important in a variety of applications, i.e., photovoltaics, solar cells and transparent substrate-based devices. Thus, the investigation of temperature impact on the physical properties of ITO layers at various atmospheres is useful for practical aspects [6]. It was shown that heat treatment of ITO thin films in air at temperatures varying from 523 K to 923 K resulted in high transparency in visible and infrared light with a minimum value of electrical resistivity of 0.7 Ω·cm [7]. Similarly, a heat treatment of DC magnetron-sputtered ITO layers in air and vacuum resulted in better crystallization and a lower value of the sheet resistance [8,9]. The annealing in various atmospheres of silver based ITO/metal/ITO structures to improve optical and electrical properties was reported in [10]. The sheet resistance of samples annealed in vacuum, oxygen, nitrogen and air at temperatures ranging from RT to 400 °C decreased to 2.45–2.53 Ω/sq. at 300 °C, while a sharp increase was measured for 500 °C, especially for samples annealed in air and oxygen. An example of influence of deposition process parameters on the microstructure and electrical properties of ITO thin films was studied in [11]. The maximum in the mobility value equal to 32 cm^2^·V^−1^·s^−1^ at carrier concentration of 2.2 × 10^20^ cm^−3^ was measured for sample deposited at an O_2_ ratio of 1.9%.

The transparent amorphous oxides are promising TE materials because the thermal conductivity can be quite low and the electrical conductivity and carrier mobility are high compared to crystalline semiconductors. The ideal thermoelectric material should demonstrate the electron crystal and phonon-glass properties with the electrical resistivity of 10^3^ Ω·cm, carrier concentration between 10^19^ and 10^21^ cm^−3^ and thermal conductivity of about 1 W·m^−1^·K^−1^ [12]. The thermal and electrical properties of thin films are morphology dependent quantities. They can be modified by changing parameters of a technological process or heat treatment in various atmospheres [13]. The ITO and In doped ZnO (IZO) were investigated to determine thermal properties. The thermal conductivity was equal to 1.5 W·m^−1^·K^−1^ for ITO layer and 1.6 W·m^−1^·K^−1^ for IZO thin film. The dependence of thermal, optical and electrical properties of polycrystalline ITO layers on deposition process parameters was reported in [14]. The thermal conductivity of ITO films varied in the range (5.86–4.00) W·m^−1^·K^−1^, and the resistivity was in the range of 0.4–3.2 mΩ·cm for O_2_ flow ratios varying in range (0–5)% during deposition.

In this work, the impact of annealing in different atmospheres on the structure, optical, thermal and thermoelectric properties of ITO thin films was investigated. Based on the thermal and electrical conductivity and the Seebeck coefficient the figure of merit was determined. The aim was to choose the optimal annealing atmosphere to improve the thermoelectric properties of ITO thin films.

## 2. Materials and Methods

A set of commercial ITO-coated glass samples (Hoya, Tokyo, Japan) with a coating thickness of 170 nm was investigated in this work. Samples were kept for 60 min. under different post-processing conditions. Annealing temperatures were equal to 200, 300 and 400 °C, and atmospheres were: air, O_2_ (nominal purity of 97%), N_2_, 10% Nitrogen-Hydrogen Mixture (NHM), CO_2_ and vacuum. The heating and cooling rates were about 3 °C·min^−1^. The X-ray diffraction (XRD) spectra were recorded on Rigaku Ultima II Max (Rigaku, Tokyo, Japan) using Cu Kα radiation.

As part of the study, atomic force microscopy (AFM) measurements were performed to determine the surface topography parameters. Topographical scans were recorded using the XE-70 model (Park Systems Inc., Suwon, Republic of Korea) working in a non-contact mode. The Budget Sensors SiNi Tap300Al-G probes with resonant frequency 300 kHz, force constant 40 N·m^−1^) were applied in topography measurements. Samples were cleaned with acetone, rinsed with deionized water and dried before experiments. The microscopic measurements allowed to determine the influence of annealing parameters, such as temperature and gas atmosphere, on the surface morphology of the layers. The thermal conductivity measurements were carried out utilizing the KNT-SThM-1a thermal probe (Kelvin NanoTechnology, Glasgow, UK). The set up was the same AFM microscope equipped with a scanning thermal microscopy (SThM) module controlling thermal probe parameters. The probe was driven by 1.5 mA DC current with a small AC component superimposed on the signal at 2.3 kHz. The amplitude of the AC component was 0.075 mA. The probe calibration was based on SThM signal measurement carried out for a set of test samples of known thermal conductivities: PMMA—0.2 W·m^−1^ K^−1^, glass—1.1 W·m^−1^ K^−1^, fused silica—1.4 W·m^−1^ K^−1^, glassy carbon—6.3 W·m^−1^ K^−1^ and silicon carbide—490 W·m^−1^ K^−1^. Before thermal measurements, topography images were recorded with an SThM probe to define surface topography parameters. Then a smooth surface area with no visible defects or contamination was selected for thermal measurements. The test samples used for probe calibration were selected to meet the criteria of a smooth glass-type surface with a roughness not exceeding several nm. The thermal properties of test samples were estimated according to the database [15,16]. The sensitivity of probe resistance measurements of SThM signal was very high due to application of lock-in amplifier (SR-830, Stanford Research Systems); the measurement uncertainties were of the order of 1–5%. The values of thermal conductivity of thin films were read from the calibration curve. The uncertainties of the SThM signal measured for test samples and ITO thin films were calculated as a standard uncertainty combining the uncertainty of logistic curve fitting and the number of repeated measurements. Determination of the thermal conductivity was based on the procedure of measuring the thermal signal at few points on the sample surface in the chosen area. The measurement in each particular point was repeated several times, and the average value of the SThM signal was calculated. The final value of the signal for a particular sample was calculated by averaging signal from all measured points. The thermal conductivity uncertainties were in the range 5–20%.

The scanning thermal microscopy allows to investigate the thermal transport phenomena with high spatial resolution [17,18,19,20]. The method was successfully applied for determination of the thermal conductivity of thin films [21,22,23].

The sheet resistivity of ITO samples was measured using a conventional four probe setup (Keithley 4200-SCS, Tektronix, Inc., Beaverton, OR, USA). The carrier concentration and the Hall mobility of the samples annealed in a range from RT to 400 °C were measured with the Temperature Dependent Hall (TDH) method in van der Pauw configuration using a magnetic field strength of 10 kG (LakeShore 7604, Lake Shore Cryotronic, Inc., Westerville, OH, USA). The Seebeck voltages were measured as a function of the applied temperature difference Δ*T* between two electrical contacts of the sample placed on a cold (cooler) and hot (heater) surface. Their temperatures, respectively *T_C_* and *T_h_*, were determined by the flow of water regulated by refrigerating circulating baths (HAAKE AC 200 with G 50, Thermo Fisher Scientific, Newington, NH, USA). The actual temperature values were measured with Pt1000 sensors attached to both sample contacts. The temperature difference Δ*T* was calculated from measured data for *T_h_* and *T_C_*. The potential difference Δ*V* was measured using Keithley 2182A voltmeter. The registered voltages Δ*V* were proportional to the temperature difference Δ*T* in accordance with the following relationship [24,25]:(1)$$∆V=S\cdot \left(T_{h}-T_{C}\right)=S\cdot ∆T$$
where *S* is the Seebeck coefficient of the measured sample.

The transmittance and reflectance spectra of ITO films were measured at room temperature utilizing an UV-Vis AvaSpec-ULS2048LTEC Spectrophotometer (Avantes, Apeldoorn, The Netherlands) with a deuterium halogen AvaLight-DH-S-BAL (Avantes) light source. Measurements were conducted across the wavelength range of 200–1100 nm.

## 3. Results

### 3.1. Structural Investigations

The XRD spectra recorded for ITO thin films annealed in different atmospheres at 400 °C are depicted in Figure 1. The XRD patterns consisted of (211), (222), (332), (431), (440), (433) and (444) peaks located at 21.5°, 30.5°, 41.8°, 45.5°, 50.8°, 52.4° and 63.5°, indicating a cubic In_2_O_3_ phase. All layers were crystalline, with a sharp (222) peak showing the preferred orientation of the ITO films.

Analysis of dominant reflection from the (222) plane allowed to determine two components, one located at 30.5° recognized as the cubic In_2_O_3_ phase and the other located at 30.4°, indicating the presence of the In_4_Sn_3_O_12_ compound. The peak was deconvoluted into In_4_Sn_3_O_12_ and In_2_O_3_ components, as shown in Figure 1b. The temperature impact on the crystallinity of layers was analyzed according to variations in the intensity of the main diffraction peaks corresponding to reflection from the (222) plane. Figure 2 shows the evolution of (222) peak components of In_2_O_3_ and In_4_Sn_3_O_12_ phases for layers annealed in different temperatures and various atmospheres. The (222) In_2_O_3_ peak intensity, (222) peak intensity ratio’s In_2_O_3_/In_4_Sn_3_O_12_, and full width at half maximum (FWHM) of (222) In_4_Sn_3_O_12_ and (222) In_2_O_3_ were considered (Figure 2a–d).

The preferential (222) orientation of ITO films decreases with the increase of annealing temperature for all atmospheric conditions (Figure 2a). A change in peak areas’ ratio can be related to the change of In_2_O_3_ and In_4_Sn_3_O_12_ phases’ share, while the FWHM is related to the phase’s crystallinity. The initial annealing at 200 °C under different atmospheres, except air, promoted the polycrystalline nature of the In_2_O_3_ phase with decreased preferential (222) orientation (Figure 2a). Further, the increase in annealing temperature decreases the preferential (222) orientation.

Low temperature annealing (200 °C) in all atmospheres except air decreased the intensities ratio, while at 400 °C in oxygen and carbon dioxide, it led to a restoration of the initial intensities ratio (2.7–2.9). A significant increase of In_4_Sn_3_O_12_ phase share for samples annealed at 400 °C in air, nitrogen and NHM can be observed (ratio of In_2_O_3_/In_4_Sn_3_O_12_~2), while the value for annealing in vacuum is in the middle of this range (Figure 2b). An increase in the intensities ratio’s suggests decomposition of the In_4_Sn_3_O_12_ phase into Sn doped In_2_O_3_, which could result in more donor carriers from substitutional Sn^4+^ on the In^3+^ lattice (Sn_In_). The grain size for two phases, In_4_Sn_3_O_12_ and In_2_O_3_, increases with temperature for all annealing atmospheres with decreasing FWHM, as shown in Figure 2c and d, respectively. The average grain size of reference ITO was estimated to be ~56 nm; annealing in vacuum had little effect on the average grain size (~57 nm), while nitrogen annealing at 400 °C increased the average grain size to ~63 nm. Similar observations were reported in [26] for RF magnetron-sputtered ITO thin films post-annealed in oxygen and nitrogen. It has been found that annealing in oxygen resulted in poor electrical properties, while annealing in nitrogen enhanced Sn doping, improved the crystallinity with larger grain size, and led to significantly improved optical and electrical properties. That effect could be attributed to the effective suppression of oxygen incorporation into the layer, maintaining oxygen vacancies in the ITO thin film.

### 3.2. Microscopic Investigations

The surface topography of ITO samples was recorded with a standard atomic force microscope (AFM) working in non-contact mode. For quantitative analysis, the built-in Gwyddion algorithms were used. The topography images, the mean surface roughness parameter (*R_a_*), and histograms of the mean surface grain size of ITO thin films annealed in different atmospheres at 400 °C are gathered in Figure 3. The impact of the annealing atmosphere on surface topography was reflected in layers’ roughness. The *R_a_* value is between 1.6 and 2.0 nm for samples annealed in air, oxygen and nitrogen, while for layers treated in vacuum, NHM and carbon dioxide, the value is about 1 nm, compared to 1.9 nm for the reference sample. Annealing in oxidizing atmospheres and nitrogen resulted in changes in surface grain size and thus mean surface grain size distribution. The grains were larger; the mean surface grain size was about (9–10) nm. Annealing in vacuum, NHM and carbon dioxide caused the decrease in the mean surface grain size. The grains were smaller and densely distributed on the surface. The mean surface grain size was in the range (5–6) nm for vacuum and NHM and (6–7) nm for CO_2_. The atmospheric impact was also reflected in the shape of grains. The elongated grains were observed on topography images recorded for samples post processed in air and nitrogen, while treatment in oxygen atmosphere resulted in large, rounded grains. Also, the surface grain size is smaller compared to bulk XRD grain size (Figure 2 and Figure 3). Both changes in surface roughness and the surface grain size could be due to variations in defects in oxygen sublattice (oxygen vacancies, V_o_ and oxygen interstitial O_I_) and oxidation states of tin (Sn^2+^ and Sn^4+^), as studied in [27].

### 3.3. Optical Properties

Transmittance and reflectance spectra of the ITO thin films were used to determine their optical band gaps. The values of the optical band gap of the ITO films annealed in various atmospheres were calculated using Tauc’s relation [28,29].

The absorption coefficient (α) for the direct allowed transition can be defined by the following Tauc’s relation:(2)$$T=\left(1-R\right)e^{\left(-\alpha d\right)}$$
(3)$$\alpha hv=A{\left(hv-E_{g}\right)}^{1/2}$$
where *T*, *R*, *d*, *hv*, *α*, *A* and *E_g_* are the transmittance, reflectance, film thickness, photon energy, absorption coefficient, a constant and optical band gap, respectively.

Using Equation (3), the absorption coefficient can be calculated from the transmittance and reflectance spectra. Figure 4a shows the curves (αh*v*)^2^ as a function of h*v*, calculated from experimental results for the transmittance and reflectance spectra. The linear relationship between (*αhv*)^2^ and *hv*, as shown in Equation (3), facilitates the determination of the optical band gap (*E_g_*) by extrapolating the linear region of each curve to zero absorption.

The ITO thin films show significant changes in the optical band gap in range from 3.93 eV to 3.99 eV for annealing (400 °C) in various atmospheres. The annealing in vacuum and oxidizing atmospheres resulted in a drop in the optical bandgap values. The absorption spectra and the energy band gap dependencies on the type of annealing atmosphere are shown in Figure 4b).

### 3.4. Thermal Properties

The thermal conductivity of ITO thin films was determined with the SThM method applying resistive thermal probes. The static and dynamic resistance of the probe was measured for probe in contact with the sample and for probe lifted over the sample surface. A detailed description of the measuring procedure can be found in Refs. [30,31]. The thermal conductivity of the sample was related to the ratio of the dynamic *R_d_* and static *R_s_* resistance difference, measured in contact with the sample and in air, called thermal signal (SThM):(4)$$SThM=\frac{{\left.\left(R_{d}-R_{s}\right)\right|}_{contact}}{{\left.\left(R_{d}-R_{s}\right)\right|}_{air}}$$

The thermal probe was calibrated using a set of test materials. The SThM signal dependence on the thermal conductivity was fitted with logistic interpolation, and the thermal conductivities of ITO samples were read from the calibration curve, as shown in Figure 5. The values were apparent thermal conductivities of the thin film/substrate system. The apparent thermal conductivity of the system *k*_s_ was correlated with the spreading resistance *R_spread_*, as:(5)$$R_{spread}=\frac{1}{4a\kappa_{s}}$$

Then, values of thermal conductivity of ITO layers were corrected to the substrate influence, assuming the layer thickness *d* = 170 nm, the contact radius *a* = 100 nm and the thermal conductivity of the substrate *k*_2_ = 1.1 W·m^−1^·K^−1^ [32]. The results for the thermal conductivity of ITO thin films are gathered in Table 1. The uncertainties of thermal signal measurements were of the order of a few percent; however, determination of the thermal conductivity was based on probe calibration. The thermal conductivity data uncertainties calculated as a standard uncertainty combining the uncertainty of fitting and the number of repeated measurements were in the range of 10–20%. The thermal conductivities of ITO thin films were in the range of 3.5–11.8 W·m^−1^ K^−1^ compared to 5.1 W·m^−1^ K^−1^ for the reference ITO sample. The post processing in different atmospheres and vacuum, except carbon dioxide, resulted in larger thermal conductivities compared to the reference sample. Samples annealed in air and nitrogen exhibit higher thermal conductivities of 10.6 and 11.8 W·m^−1^ K^−1^, respectively. Sample treated in carbon dioxide shows the lowest value of 3.5 W·m^−1^ K^−1^. From XRD analysis, samples annealed in air, nitrogen and NHM revealed a higher In_4_Sn_3_O_12_ phase contribution and an increase in the thermal conductivity. Annealing in vacuum at 400 °C resulted in the thermal conductivity value of 8.3 W·m^−1^ K^−1^, placing it in the middle range of ITO thermal conductivities, following the trend observed in XRD analysis. Annealing in oxidizing atmospheres (air, O_2_) decreased V_o_ and increased O_I_, while reducing atmospheres, such as NHM, increased V_o_ and decreased O_I_, affecting the carrier concentration.

Heat conduction in semiconducting materials comes from two mechanisms: the carrier (electrons and holes) *k_e_* conduction and the lattice one (phonons) *k_l_*. The carrier’s contribution to thermal conductivity is related to the electrical conductivity through the Wiedemann-Franz law. Typically the *k_e_* is determined from electrical conductivity measurements, and the lattice thermal conductivity is calculated as a difference between total thermal conductivity and electron part *k_e_*. Ashida et al. investigated thermal transport properties in polycrystalline ITO thin films with carrier concentrations in the range of 1.9 × 10^20^–1.2 × 10^21^ cm^−3^ and demonstrated metal-like behavior showing linear dependency between the thermal and electrical conductivities [14]. The reduction in thermal conductivity can be achieved through the scattering of phonons in the material, for example, by the creation of point defects such as interstitial atoms, vacancies or scattering phonons at interfaces in nanostructured layered materials.

### 3.5. Electrical and Thermoelectrical Properties

The electrical conductivity, carrier concentration and mobility of ITO layers determined from Hall measurements are depicted in Figure 6a–c. The electrical conductivity of ITO samples decreased with temperature above 300 °C, with a significant drop in values of 60% for layers annealed in oxygen and about 45% for layers annealed in air and vacuum at 400 °C. Annealing in N_2_ atmospheres from RT to 400 °C resulted in a decrease of a few percent in the electrical conductivity, while annealing at 400 °C in NHM resulted in 10% increase. The CO_2_ annealing retained the electrical conductivity value. The inert atmospheres such as N_2_, NHM and CO_2_ retained the carrier concentration (Figure 6b).

The NHM-reducing atmosphere increased the carrier concentration, while the oxidizing atmospheres (air and O_2_) decreased it, suggesting the increase of donor V_o_ or metal clusters [33] and the decrease of V_o_, respectively. Annealing in vacuum caused the segregation of Sn at the grain boundary, reducing the carrier concentration [34]. Surprisingly, annealing at 400 °C in CO_2_ increased the mobility (~37 cm^2^·V^−1^·s^−1^), while other atmospheres decreased the mobility (~29–33 cm^2^·V^−1^·s^−1^) compared to 35.7 cm^2^·V^−1^·s^−1^ for the reference ITO sample (Figure 6c). The ITO layers annealed at 400 °C in N_2_ and NHM exhibit a larger value of carrier concentration compared to the reference ITO sample. Similarly, the improvement of electrical properties of ITO thin films while annealing in nitrogen atmospheres was reported in [35]. It was shown that the lowest value of the electrical resistivity equal to 0.23 mΩ·cm was determined for layers annealed at 400 °C in nitrogen. The carrier concentration and mobility of N_2_-annealed ITO layers increase with temperature increase. The authors suggested that some electrons are trapped in small areas in the amorphous structure of films. The structure became crystalline with temperature increase, and these electrons would be released from the trap, which contributes to the increase of carrier concentration.

The transparent oxide thin films can be promising TE materials because their electrical conductivity and carrier mobility are high compared to crystalline semiconductors. The ZT parameter is sensitive to the thermal conductivity value. The thermal conductivity can be decreased by doping or deposition of amorphous layers [36].

The potential difference Δ*V* versus time with applied temperature difference Δ*T* for all ITO samples was registered to determine the Seebeck coefficients (S). Based on the experimental data, the calculations of the Seebeck coefficients and their uncertainty were performed by fitting the results with linear function (Equation (1)). Additionally, based on the thermal and electrical conductivity measurements, the ZT parameter was determined (Table 1). Figure 7a presents the Seebeck coefficient and thermal conductivity versus type of annealing atmosphere. Similarly, the results for the Seebeck coefficient and electrical conductivity versus type of annealing atmosphere are depicted in Figure 7b. Interestingly, the S values do not follow the trends in carrier concentration but are correlated to *R* values, as demonstrated in [13]. The annealing in air (78% N_2_ + 21% O_2_) resulted in the highest S parameters. Similarly, annealing in 99% O_2_ and the vacuum increased the S value compared to the reference sample. On the contrary, annealing N_2_, CO_2_ and NHM retained the S value unchanged. The Seebeck coefficient follows the trend observed in the thermal conductivity, with an exception for the N_2_ annealed sample (Figure 7a). The electrical conductivity values change against the trend in S value. The drop in electrical conductivity is observed for samples annealed in vacuum and oxidizing atmospheres, while annealing N_2_, CO_2_ and NHM improved the value (Figure 7b).

## 4. Discussion

The investigated ITO samples were polycrystalline layers of good quality. The annealing in various atmospheres and vacuums resulted in differences in the structure, optical, thermal and thermoelectrical properties of thin films. Post-annealing from 200 °C to 400 °C resulted in slight variations in the mean surface roughness parameter. The value of 2.0 nm was determined for samples annealed in air, O_2_ and N_2_, while for layers treated in vacuum, NHM and CO_2_ the value was two times less. Structure analysis revealed a significant increase of In_4_Sn_3_O_12_ phase share for samples annealed at 400 °C in air, N_2_ and NHM, with a ratio of In_2_O_3_/In_4_Sn_3_O_12_~2. It was shown from FWHM analysis that the grain size for two phases, In_4_Sn_3_O_12_ and In_2_O_3_, increases with temperature for all annealing atmospheres. The annealing in vacuum and oxidizing atmospheres resulted in a drop of the optical bandgap values, while N_2_, NHM and CO_2_ atmospheres effected in comparable values of the optical bandgap as reference ITO. The different atmospheres annealing influenced the thermal and electrical conductivities of ITO thin films. The highest thermal conductivity, equal to 12 W·m^−1^ K^−1^, was determined for samples annealed in air and N_2_, while the lowest value, 3.5 W·m^−1^ K^−1^, was measured for samples annealed in CO_2_. Changes in thermal conductivity were correlated with surface topography parameters; the highest values of surface mean grain size were obtained for layers annealed in air and N_2_, and the lowest for CO_2_ and NHM. It was reported that grain size is dependent on layer thickness and deposition process parameters [37]. Smaller grains dominated in the ITO samples post processed in CO_2_, NHM and vacuum, whereas grains were larger while annealed in oxidizing atmospheres (air and O_2_) and N_2_. The heat transport in thin films is limited by structural defects and the quality of the crystal lattice. In polycrystalline layers, an increase in thermal conductivity is typically observed with an increase in the average grain size.

Value changes in thermoelectrical properties were dependent on type of annealing atmosphere. The drop in electrical conductivity and Seebeck coefficient was observed for samples treated in oxidizing atmospheres and vacuum, while application of NHM, N_2_ and CO_2_ resulted in values comparable with those of the reference ITO sample. The annealing in the NHM-reducing atmosphere could result in the creation of V_o_ and metal In clusters. It was reported that for temperatures higher than 350 °C metallic indium or indium/tin were extracted, which correlated with a decrease in electrical conductivity and transparency. The further increase in annealing temperature was favorable for the tendency of metallic indium to create larger aggregates and microcrack formation during sintering [33]. Similarly for ITO thin films, the drop in electrical properties was noticeable for temperatures above 300 °C in vacuum and oxidizing atmospheres. The optimal properties towards TE applications, i.e., the low thermal conductivity and high electrical conductivity, were determined for ITO annealed in CO_2_ and NHM atmospheres. The results are in good agreement values obtained for RF sputtered ITO thin films [38]. Summarizing, the CO_2_ seems to be an optimal atmosphere to obtain good TE generator parameters—high ZT. The maximum of the *ZT* parameter corresponds to a high value of the Seebeck coefficient and is sensitive to the thermal conductivity value. The comparison of the ZT values obtained in this work with literature data is depicted in Figure 8.

## 5. Conclusions

Basing on the complex characterization of ITO thin films, it was shown that annealing in oxidizing atmospheres such as air or O_2_ deteriorated the quality of the layers and their optical and thermoelectrical properties. On the contrary, heat treatment in CO_2_ and NHM promoted crystallization as well as improved thermal and thermoelectrical properties of ITO layers. We also performed a systematic analysis of temperature impact on the surface morphology and structure of ITO thin films. The mean grain size increases with temperature for all samples, and the changes in structure were more pronounced for layers exposed to oxidizing atmospheres, such as air and O_2_. The drop in electrical conductivity and carrier concentration was observed for temperatures higher than 350 °C while annealing in air, O_2_ and vacuum. The improvement of the electrical properties was observed for layers annealed up to 400 °C in CO_2_ and NHM atmospheres. Thus, carbon dioxide and nitrogen-hydrogen mixtures seem to be the optimal post annealing atmospheres to improve thermoelectrical properties. The highest value of the ZT parameter was determined for samples annealed in CO_2_ atmosphere. The general conclusion drawn from our work is that annealing in different atmospheres allows for controlled changes in the structure and physical properties of ITO layers.

## Figures and Tables

**Figure 1 materials-17-04606-f001:**
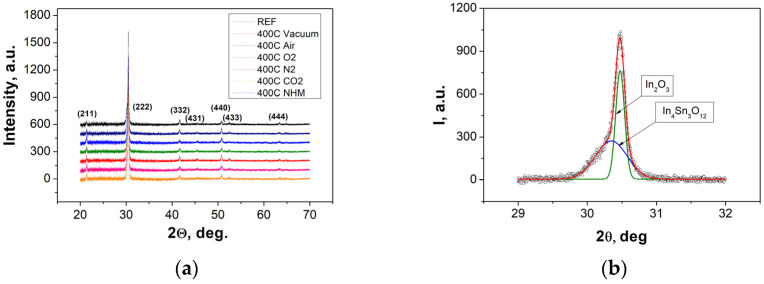
XRD spectra (**a**) recorded for ITO thin films annealed in different atmospheres at 400 °C and (**b**) deconvoluted (222) peak for reference sample.

**Figure 2 materials-17-04606-f002:**
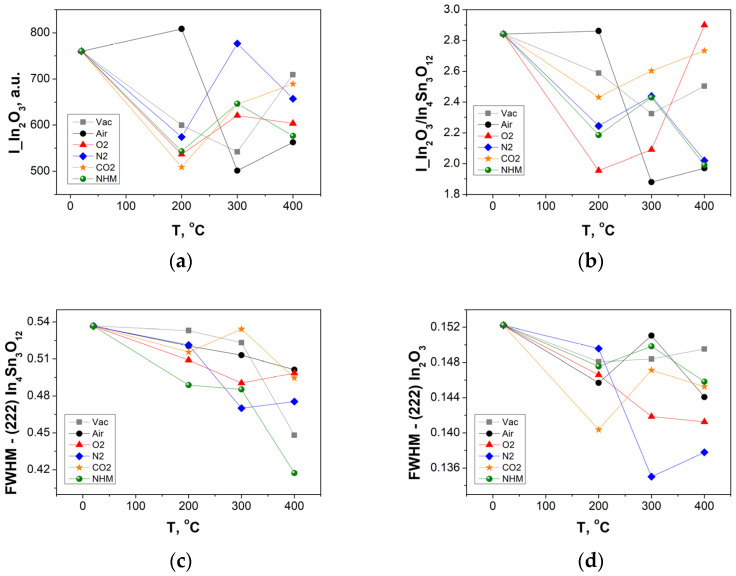
(**a**) The main diffraction peak (222) decomposed into two In_4_Sn_3_O_12_ to In_2_O_3_ components, (**b**) two components area ratio for ITO layers annealed from RT to 400 °C in various atmospheres, (**c**) FWHM factor of (222) reflection recorded as a function of annealing temperature: the In_4_Sn_3_O_12_ component and (**d**) the In_2_O_3_ component.

**Figure 3 materials-17-04606-f003:**
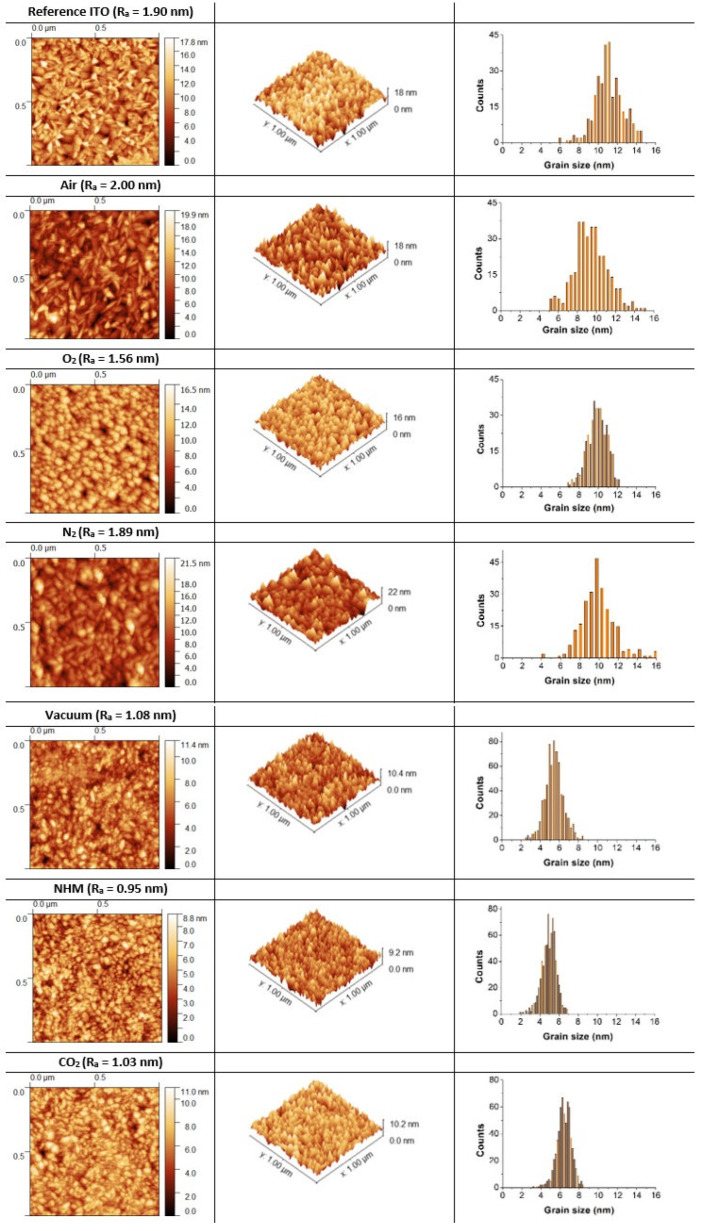
1 × 1 mm^2^ AFM topographic images with 3D view of ITO thin films annealed at 400 °C in various atmospheres and histogram of surface mean grain size analysis.

**Figure 4 materials-17-04606-f004:**
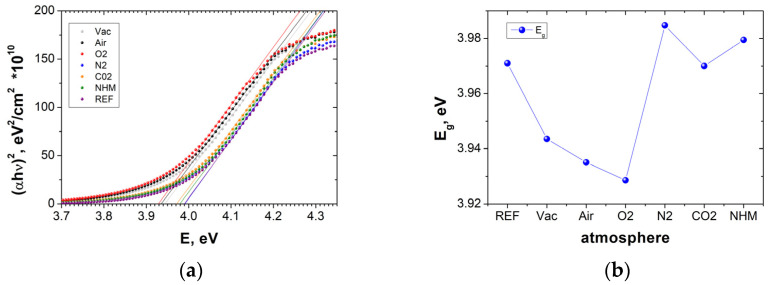
Absorption spectra (**a**), energy band gap (**b**) of ITO thin films annealed in different atmospheres at 400 °C.

**Figure 5 materials-17-04606-f005:**
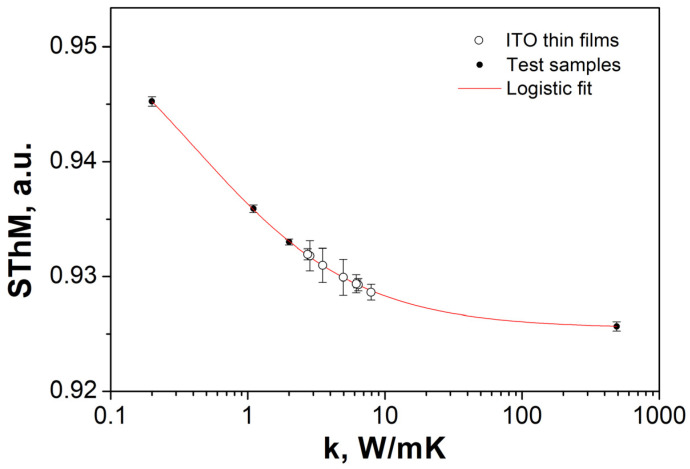
Calibration curve for thermal measurements.

**Figure 6 materials-17-04606-f006:**
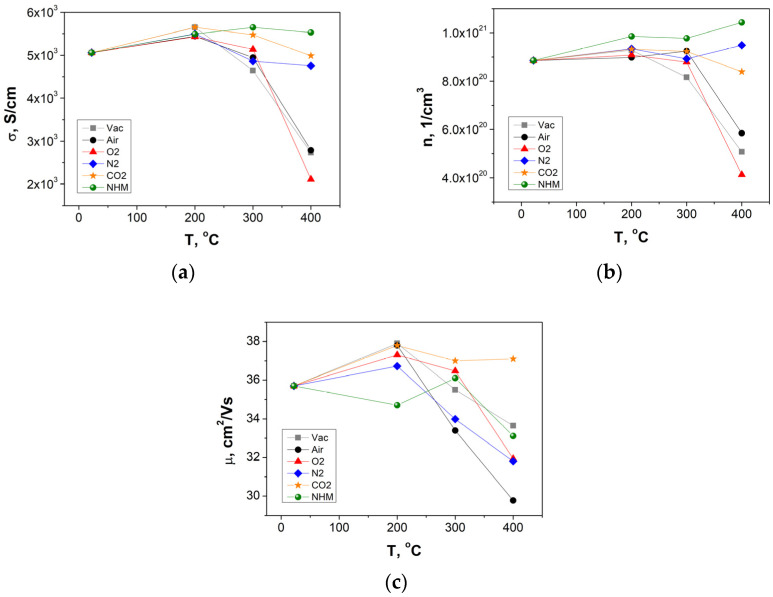
Electrical conductivity (**a**), carrier concentration (**b**), mobility (**c**) as a function of the temperature of ITO thin films annealed in different atmospheres.

**Figure 7 materials-17-04606-f007:**
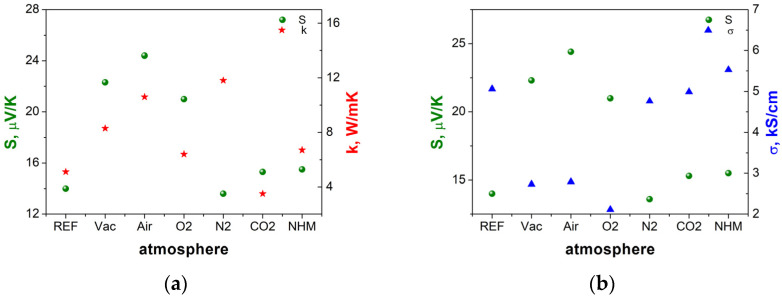
Seebeck coefficient and thermal conductivity (**a**), Seebeck coefficient and electrical conductivity (**b**) versus type of annealing atmosphere.

**Figure 8 materials-17-04606-f008:**
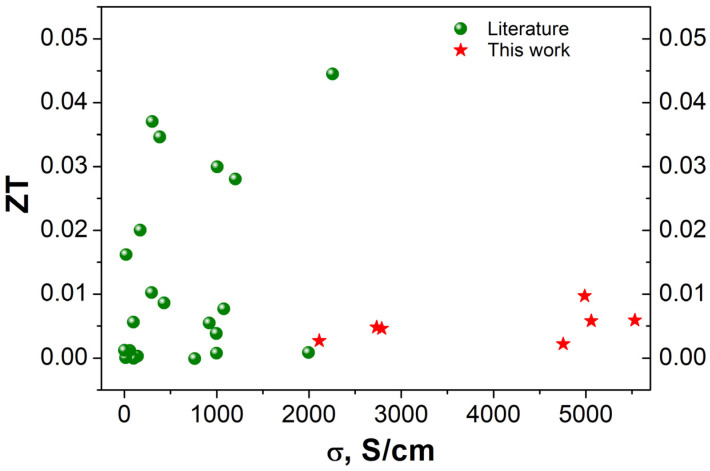
Comparison of figure of merit ZT as a function of electrical conductivity in this work with results collected by Kim, S, et al. [13].

**Table 1 materials-17-04606-t001:** Topographical, optical, thermal and thermoelectrical properties of ITO samples annealed in different atmospheres at 400 °C.

Atmosphere	R_a_, nm	E_g_, eV	k, W·m^−1^ K^−1^	σ, 10^3^ S·cm^−1^	S, μV·K^−1^	ZT, 10^−3^
REF	1.90	3.97	5.1	5.06	14.1	5.8
Vacuum	1.08	3.94	8.3	2.73	22.3	4.8
Air	2.00	3.93	10.6	2.79	24.4	4.6
O_2_	1.56	3.93	6.4	2.11	20.9	2.7
N_2_	1.89	3.98	11.8	4.76	13.6	2.2
CO_2_	1.03	3.97	3.5	4.99	15.3	9.7
NHM	0.95	3.98	6.7	5.53	15.5	5.9

## Data Availability

The data presented in this study are available on request from the corresponding author.

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
