# Peer review of "Impact of Annealing in Various Atmospheres on Characteristics of Tin-Doped Indium Oxide Layers towards Thermoelectric Applications"

_materials, 2024, doi:10.3390/ma17184606_

Round 1

Reviewer 1 Report

Comments and Suggestions for Authors

The manuscript describes a multi-method analysis of ITO layers with annealing in different atmospheric conditions and temperature ranges. Indeed, this analysis goes beyond already reported results as it systematically addresses the influence of the different factors with regard to structural, surface and electrical changes. 

A few things should be revised before the manuscript is acceptable for publication: 

1. The introduction part is somehow puzzling to the reader and should be rewritten: In the current version, it tries to summarize several selected older publications, however, it should more focus on the general aspects than give detailed quantitative former results. The comparison of already existent results should better be part of the discussion section. The authors should avoid global statements in particular when not supported by citations (e.g., easily found in many [...] papers,  weakly/less investigated, etc.) as well as trivial statements like "analysis of anneling temperature [...] ist useful to investigate the thermal stability [...]". 

2. In the Materials and Methods section, the authors describe a set of test samples with known thermal conductivities and a calibration curve, but without reference. In particular, the quality of the calibration in the sence of its uncertainties should be given. Generally, the authors should add some discussion of the uncertainties of their measurements: In fig 4, some uncertainties are presented for the ITO films calibration, but basically a uncertainty estimate should be added for all measurands, and also be shown or mentioned in result figures.

The notation for the temperature unit should be the same in Figs. 2 and 5 (deg C or °C).

3. The discussion of the results remains only  rudimentary regarding the physical processes: E.g., in the introduction (line 52) it is mentioned that trapping of electrons is assumed as  a cause for resistivity change. Can the present results be interpreted in that way? What could be an explanation for the larger grain size in N2, as it is definitely not oxidizing? Furthermore, a few words should be dropped to motiviate the discussion in terms of the ZT  parameter.

Overall, the manuscript gives a extensive description of the measurement methods and results, but should be revised  towards a more coherently discussion of previous data and the microscopic model behind.

Author Response

Reply: We are grateful for the reviewer's suggestion and for careful reading of our manuscript. We checked the manuscript files again and corrected it according reviewer's comments.

The manuscript describes a multi-method analysis of ITO layers with annealing in different atmospheric conditions and temperature ranges. Indeed, this analysis goes beyond already reported results as it systematically addresses the influence of the different factors with regard to structural, surface and electrical changes. 

A few things should be revised before the manuscript is acceptable for publication: 

  1. The introduction part is somehow puzzling to the reader and should be rewritten: In the current version, it tries to summarize several selected older publications, however, it should more focus on the general aspects than give detailed quantitative former results. The comparison of already existent results should better be part of the discussion section. The authors should avoid global statements in particular when not supported by citations (e.g., easily found in many [...] papers,  weakly/less investigated, etc.) as well as trivial statements like "analysis of anneling temperature [...] ist useful to investigate the thermal stability [...]". 

Reply: We are grateful for the reviewer's suggestions. We have revised and rewritten the Introduction section. We also moved a part of already existent results from Introduction to the discussion section: 3.4 Electrical and thermoelectrical properties.

The electrical properties of ITO thin films were improved while annealing nitrogen atmospheres [[i]]. The lowest value of the electrical resistivity equal to 0.23 mW·cm was determined for layers annealed at 400°C in nitrogen atmosphere. The carriers concentration and mobility of ITO layers annealed in N2 atmosphere increased with temperature increase. Authors suggested that some electrons are trapped in small areas in the amorphous structure of films. It was shown that structure became crystalline with temperature increase and these electrons would be released from the trap what contributes to the increase of carrier concentration. Similarly, the ITO layers annealled at 400 °C in nitrogen and hydrogen-nitrogen mixture exhibit largere value of carriers concentration compared to reference ITO sample (Figure 6 (b)).

Reply: we moved also the section 3.5 Optical properties right after the part about 3.2 Microscopic investigations to improve the logical sequence of the manuscript, thermal properties now follow electrical and thermoelectric ones. We have also changed Figure 7 to be more consistent with the previous graphs. Figure 7 shows now the comparison of the Seebeck coefficient with the thermal conductivity and electrical conductivity as a function of different annealing atmospheres.

  1. In the Materials and Methods section, the authors describe a set of test samples with known thermal conductivities and a calibration curve, but without reference. In particular, the quality of the calibration in the sense of its uncertainties should be given. Generally, the authors should add some discussion of the uncertainties of their measurements: In fig 4, some uncertainties are presented for the ITO films calibration, but basically a uncertainty estimate should be added for all measurands, and also be shown or mentioned in result figures.

Reply: We have revised and substantially expanded the Materials and Methods sections of the manuscript and improved the results presented in the Fig.5 (old Fig. 4), as requested by the reviewer. In the previous version, the measurement uncertainties were marked for all measurement points, but the markers were too large and obscured the uncertainties. Figure 5 has been changed, i.e. the marker size has been reduced. We have also added a discussion on measurement uncertainties in the text as below:

Before thermal measurements topography images were recorded with SThM probe to define surface topography parameters. Then a smooth surface area with no visible defects or contamination was selected for thermal measurements. The test samples used for probe calibration were selected to meet the criteria of a smooth glass-type surface with a roughness not exceeding several nm. The thermal properties of test samples were estimated according to the database [[ii],[iii]]. The sensitivity of probe resistance measurements of SThM signal was very high due to application of lock-in amplifier (SR-830, Stanford Research Systems), the measurement uncertainties were of order of 1-5%. The values of thermal conductivity of thin films were read from calibration curve. The uncertainties of SThM signal measured for test samples and ITO thin films were calculated as a standard uncertainty combining the uncertainty of logistic curve fitting and the number of repeated measurements. Determination of the thermal conductivity was based on procedure of measuring thermal signal in few points on sample surface in chosen area. The measurement in each particular point was repeated 5 times and average value of SThM signal was calculated. The final value of the signal for particular sample was calculated by averaging signal from all measured points. The thermal conductivity uncertainties were in the range 5-20%.

The notation for the temperature unit should be the same in Figs. 2 and 5 (deg C or °C).

Reply: We have revised and corrected the units in Fig.2 and 6 (Figs. 2 and 5 in previous version)

  1. The discussion of the results remains only  rudimentary regarding the physical processes: E.g., in the introduction (line 52) it is mentioned that trapping of electrons is assumed as  a cause for resistivity change. Can the present results be interpreted in that way? What could be an explanation for the larger grain size in N2, as it is definitely not oxidizing? Furthermore, a few words should be dropped to motiviate the discussion in terms of the ZT  parameter.

Reply: We are grateful for the reviewer's suggestions, the ref.10 and a part about “that trapping of electrons” was moved to section 3.5 and an explanation was added:

The ITO layers annealled at 400 °C in N2 and NHM exhibit larger value of carriers concentration compared to reference ITO sample. Similarly, the improvement of electrical properties of ITO thin films while annealing in nitrogen atmospheres was reported in [[iv]]. It was shown, that the lowest value of the electrical resistivity equal to 0.23 mW·cm was determined for layers annealed at 400°C in nitrogen. The carriers concentration and mobility of N2 annealed ITO layers increases with temperature increase. Authors suggested that some electrons are trapped in small areas in the amorphous structure of films. The structure became crystalline with temperature increase and these electrons would be released from the trap what contributes to the increase of carrier concentration.

Reply: In our work the mean grain size (XRD) and the mean surface grain size (AFM) for ITO layer annealed in N2 was 63 nm and (9-10) nm with Ra = 1.9 nm. Some comments and explanation for the larger grain size for layer annealed in N2 were added in point 3. Results, section 3.1 Structural investigations:

Similar observations were reported in [[v]] for RF magnetron sputtered ITO thin films post annealed in oxygen and nitrogen. It has been found that annealing in oxygen resulted in poor electrical properties, while annealing in nitrogen enhanced Sn doping, improved the crystallinity with larger grain size and led to a significantly improved optical transmittance and low electrical resistance. That result could be attributed to the combined effects of effective suppression of oxygen incorporation into films, maintaining oxygen vacancies in the ITO film.

Overall, the manuscript gives a extensive description of the measurement methods and results, but should be revised  towards a more coherently discussion of previous data and the microscopic model behind.

[i].                     Guillén, C. and Herrero, J. Structure, optical, and electrical properties of indium tin oxide thin films prepared by sputtering at room temperature and annealed in air or nitrogen. J. Appl. Phys. 2007, 101, 073514-1-7. https://doi.org/10.1063/1.2715539

[ii].             http://www.thermtest.com/material-property-search/

[iii].             Tanzi, M. C.; Farè, S.; Candiani, G. Organization, Structure, and Properties of Materials. Foundations of Biomaterials Engineering. 2019, 3-103. https://doi.org/10.1016/B978-0-08-101034-1.00001-3

[iv].                    Guillén, C. and Herrero, J. Structure, optical, and electrical properties of indium tin oxide thin films prepared by sputtering at room temperature and annealed in air or nitrogen. J. Appl. Phys. 2007, 101, 073514-1-7. https://doi.org/10.1063/1.2715539

[v].             Parida, B.; Gil, Y.; Kim, H. Highly Transparent Conducting Indium Tin Oxide Thin Films Prepared by Radio Frequency Magnetron Sputtering and Thermal Annealing. J. Nanosci. Nanotechnol. 2019, 19, 1455-1462. https://doi.org/10.1166/jnn.2019.16242

Reviewer 2 Report

Comments and Suggestions for Authors

In this paper, the indium tin oxide (ITO) layer was annealed at room temperature to 400℃ under three different atmospheres: air, nitrogen/hydrogen mixture (NHM), and oxygen. The effects of different atmospheres and temperatures on the structure, thermal, and electrical properties of the ITO layer were studied. The results show that the crystallinity remains unchanged under different atmosphere conditions, the thermal conductivity increases with the increase of temperature, and the resistivity doubles under oxidation atmosphere and vacuum treatment. The research method is appropriate, the research method is reasonable, and the language expression is smooth. The research conclusion has certain scientific value for the industry and is worthy of publication. However, there are the following deficiencies, which need to be revised.

1. The structure of the abstract of this paper is unreasonable, which is very confusing. It is not expressed in accordance with the format of the abstract, and there is no deep innovative scientific conclusion, which makes it difficult to arouse readers' interest in this paper. First of all, the scientific problems of this research should be put forward, and the end should point out the value or significance of this research to scientific research, so it needs to be rewritten;

2. Most of the introduction of this paper refers to the literature of 5 years ago or even 10 years ago, which cannot reflect the research value of this topic. It is suggested that the author refer to the literature of the recent 5 years, especially the literature of the recent 2 years, so it is necessary to revise the introduction;

3. What parts do you consider original or relevant for the field?

4. The quotation format of all references in the paper is incorrect, please revise it carefully;

5. All the graphs in the text are subgraphs such as a, b,...... It is not marked in the figure, so people can not understand, please modify it;

6. Many expressions in the text are inconsistent and incorrect, such as "Figure 179 2 (a), (b), (c), (d)) "on page 5, line 179, and "Figure 5 (a) -- (c)" on page 9, line 282, please revise and check the whole text.

7. There are too many conclusion parts in the article, please express them according to the article, and simplify the conclusion to make it readable;

8. The number of references in the paper in the recent five years is too small, especially those in the recent two years. Please add some references in the recent five years, especially those in 2023 and 2024, and please revise them.

9. What specific gap in the field does the paper address?

I hope the author can revise it carefully to improve the quality of this manuscript.

Author Response

Reply: We are grateful for the reviewer's suggestion and for careful reading of our manuscript. We checked the manuscript files again and corrected it according reviewer's comments.

Comments and Suggestions for Authors

In this paper, the indium tin oxide (ITO) layer was annealed at room temperature to 400℃ under three different atmospheres: air, nitrogen/hydrogen mixture (NHM), and oxygen. The effects of different atmospheres and temperatures on the structure, thermal, and electrical properties of the ITO layer were studied. The results show that the crystallinity remains unchanged under different atmosphere conditions, the thermal conductivity increases with the increase of temperature, and the resistivity doubles under oxidation atmosphere and vacuum treatment. The research method is appropriate, the research method is reasonable, and the language expression is smooth. The research conclusion has certain scientific value for the industry and is worthy of publication. However, there are the following deficiencies, which need to be revised.

  1. The structure of the abstract of this paper is unreasonable, which is very confusing. It is not expressed in accordance with the format of the abstract, and there is no deep innovative scientific conclusion, which makes it difficult to arouse readers' interest in this paper. First of all, the scientific problems of this research should be put forward, and the end should point out the value or significance of this research to scientific research, so it needs to be rewritten;

Reply: We are grateful for the reviewer's suggestions. We have revised and rewritten the abstract.

Abstract: The aim of this work was to investigate the possibility of modifying the physical properties of indium tin oxide (ITO) layers by annealing them in different atmospheres and temperatures. Samples were annealed in vacuum, air, oxygen, nitrogen, carbon dioxide and a mixture of nitrogen with hydrogen (NHM) in temperatures from 200 °C to 400 °C. Annealing impact on the crystal structure, optical, electrical, thermal and thermoelectric properties was examined. It has been found from XRD measurements, that for samples annealed in air, nitrogen and NHM at 400 °C the In2O3/In4Sn3O12 shares ratio decreased, resulting in a significant increase of In4Sn3O12 phase. The annealing at the highest temperature resulted also in the mean grain size increase, especially for the layer annealed in nitrogen. Annealing in air and nitrogen resulted in larger grains, while vacuum, NHM and carbon dioxide atmospheres caused the decrease in the mean surface grain size. The annealing in vacuum and oxidizing atmospheres effected in drop of optical bandgap and poor electrical properties. The carbon dioxide seems to be optimal atmosphere to obtain good TE generator parameters – high ZT. The general conclusion is that annealing in different atmospheres allows for controlled changes in the structure and physical properties of ITO layers.

  1. Most of the introduction of this paper refers to the literature of 5 years ago or even 10 years ago, which cannot reflect the research value of this topic. It is suggested that the author refer to the literature of the recent 5 years, especially the literature of the recent 2 years, so it is necessary to revise the introduction;

Reply: We have revised and rewritten the Introduction and added more recent literature, among them:

  • Ghosh, D.K.; Bose, S.; Das, G.; Mukhopadhyay, S.;Sengupta, A. Realization of performance enhancement of thin film silicon solar cells by applying ITO/AZO bilayer TCO films as front electrode. J Mater Sci: Mater Electron2023, 34, 2189. https://doi.org/10.1007/s10854-023-11570-9
  • Ohashi, N.; Kaneko, R.; Sakai, C.; Wasai, Y.; Higuchi, S.; Yazawa, K.; Tahara, H.; Handa, T.; Nakamura, T.; Murdey, R.; Kanemitsu, Y.; Wakamiya, A. Bilayer Indium Tin Oxide Electrodes for Deformation-Free Ultrathin Flexible Perovskite Solar Cells. Sol. RRL 2023, 7, 2300221. https://doi.org/10.1002/solr.202300221
  • Heffner, H.; Soldera, M.; Lasagni, A.F. Optoelectronic performance of indium tin oxide thin films structured by sub-picosecond direct laser interference patterning. Sci Rep2023, 13, 9798. https://doi.org/10.1038/s41598-023-37042-y
  • Luo, B.; Cao, L.; Gao, H.; Zhang, Zh.; Luo, F.; Zhou, H.; Ma, K.; Liu, D.; Miao, M. Superior Thermoelectric Performance of Robust Column-Layer ITO Thin Films Tuning by Profuse Interfaces. ACS Appl. Mater. Interfaces. 2022, 14, 36258-36267. https://doi.org/10.1021/acsami.2c09907
  • Tanzi, M. C.; Farè, S.; Candiani, G. Organization, Structure, and Properties of Materials. Foundations of Biomaterials Engineering. 2019, 3-103. https://doi.org/10.1016/B978-0-08-101034-1.00001-3
  • Zhang, Q.; Zhu, W.; Zhou, J.; Deng, Y. Realizing the Accurate Measurements of Thermal Conductivity over a Wide Range by Scanning Thermal Microscopy Combined with Quantitative Prediction of Thermal Contact Resistance. Small 2023, 19, 2300968-1-10. https://doi.org/10.1002/smll.202300968
  • Yang, X.; Wang, Ch.; Lu, R.; Shen, Y.; Zhao, H.; Li, J.; Li, R.; Zhang, L.; Chen, H.; Zhang, T.; Zheng, X. Progress in measurement of thermoelectric properties of micro/nano thermoelectric materials: A critical review. NanoEnergy. 2022, 101, 107553-1-21. https://doi.org/10.1016/j.nanoen.2022.107553
  • Yamashita, Y.; Honda, K.; Yagi, T.; Jia, J.; Taketoshi, N.; Shigesato, Y. Thermal conductivity of hetero-epitaxial ZnO thin films on c- and r-plane sapphire substrates: Thickness and grain size effect. Appl. Phys. 2019, 125, 035101-1-11. doi: 10.1063/1.5055266
  • Parida, B.; Gil, Y.; Kim, H. Highly Transparent Conducting Indium Tin Oxide Thin Films Prepared by Radio Frequency Magnetron Sputtering and Thermal Annealing. Nanosci. Nanotechnol. 2019, 19, 1455-1462. https://doi.org/10.1166/jnn.2019.16242
  • Tchenka, A.; Agdad, A.; Mellalou, A.; Chaik, M.; Haj, D. A.; Narjis, A.; Nkhaili, L.; Ibnouelghazi, E.; Ech‑Chamikh, E. Spectroscopic Investigations and Thermoelectric Properties of RF‑Sputtered ITO Thin Films. Electron. Mater. 2022, 51, 1401–1408. https://doi.org/10.1007/s11664-021-09416-3
  1. What parts do you consider original or relevant for the field?

Reply: Original or relevant for the field we consider the full characterization of the thermophysical properties of ITO thin films, with particular emphasis on the analysis of thermal properties. The thermal conductivity coefficient of thin films is a key parameter of thermoelectric materials. The ZT parameter is very sensitive to the thermal conductivity values. As we have shown in this work, electrical properties have a weaker effect on ZT, while reducing the thermal conductivity value significantly improved the ZT parameter.

  1. The quotation format of all references in the paper is incorrect, please revise it carefully;

Reply: The quotation format of all references in the paper was revised, corrected and expanded.

  1. All the graphs in the text are subgraphs such as a, b,...... It is not marked in the figure, so people can not understand, please modify it;

Reply: All the graphs in the text were marked according to Reviewer suggestions.

  1. Many expressions in the text are inconsistent and incorrect, such as "Figure 179 2 (a), (b), (c), (d)) "on page 5, line 179, and "Figure 5 (a) -- (c)" on page 9, line 282, please revise and check the whole text.

Reply: Most of the text in the manuscript was revised and corrected, and some sections were expanded. We moved the section 3.5 Optical properties right after the part about 3.2 Microscopic investigations to improve the logical sequence of the manuscript, thermal properties now follow electrical and thermoelectric ones. We have also changed Figure 7 to be more consistent with the previous graphs. Figure 7 shows now the comparison of the Seebeck coefficient with the thermal conductivity and electrical conductivity as a function of different annealing atmospheres.

  1. There are too many conclusion parts in the article, please express them according to the article, and simplify the conclusion to make it readable;

Reply: The conclusions section was rewritten and corrected:

  1. Conclusions

Basing on complex characterization of ITO thin films, it was shown that annealing in oxidizing atmospheres like air or O2 deteriorated the quality of the layers, their optical and thermoelectrical properties. On the contrary, heat treatment in CO2 and NHM promoted crystallization as well as improved thermal and thermoelectrical properties of ITO layers. We also performed a systematic analysis of temperature impact on surface morphology and structure of ITO thin films. The mean grain size increases with temperature for all samples and the changes in structure were more pronounced for layers exposed to oxidizing atmospheres, like air and O2. The drop in electrical conductivity and carrier concentration was observed for temperatures higher than 350 °C while annealing in air, O2 and vacuum. The improvement of the electrical properties was observed for layers annealed up to 400 °C in CO2 and NHM atmospheres. Thus, carbon dioxide and nitrogen hydrogen mixture seem to be the optimal post annealing atmospheres to improve thermoelectrical properties. The highest value of ZT parameter was determined for samples annealed in CO2 atmosphere. General conclusion drawn from our work is that that annealing in different atmospheres allows for controlled changes in the structure and physical properties of ITO layers.

  1. The number of references in the paper in the recent five years is too small, especially those in the recent two years. Please add some references in the recent five years, especially those in 2023 and 2024, and please revise them.

Reply: The references in the manuscript were revised and expanded, following Reviewer's suggestions from point 2.

  1. What specific gap in the field does the paper address?

Reply: The manuscript addresses to application of post processing to modify physical properties of ITO thin films. The aim was to choose optimal atmosphere to improve thermoelectric properties of ITO thin films. The last section of Introduction was rewritten:

In this work the impact of annealing in different atmospheres on structure, optical, thermal and thermoelectric properties of ITO thin films was investigated. Based on the thermal and electrical conductivity, and the Seebeck coefficient, the figure of merit parameter was determined. The aim was to choose the optimal annealing atmosphere to improve thermoelectric properties of ITO thin films.

I hope the author can revise it carefully to improve the quality of this manuscript.

Round 2

Reviewer 1 Report

Comments and Suggestions for Authors

The authors revised the manuscript in response to the criticism stated in the first report. The improvements are significant and definitely promote the quality of presentation and the conclusiveness of the findings.

Therefore I have no objections to the acceptance of the manuscript in its present final form.